# A graph neural network-based approach for predicting SARS-CoV-2–human protein interactions from multiview data

**Sumanta Ray**[ORCID][1,2☯*], **Syed Alberuni**[3☯], **Alexander Schönhuth**[2☯]

**1** Data Science Unit, The West Bengal National University of Juridical Sciences, Kolkata, West Bengal, India, **2** Genome Data Science, University of Bielefeld, Bielefeld, Germany, **3** Department of Computer Science and Engineering, Aliah University, Kolkata, West Bengal, India

☯ These authors contributed equally to this work.
* sumantaray@nujs.edu

**Data availability statement:** All relevant data are within the manuscript and its Supporting information files.

## Abstract

The COVID-19 pandemic has demanded urgent and accelerated action toward developing effective therapeutic strategies. Drug repurposing models (in silico) are in high demand and require accurate and reliable molecular interaction data. While experimentally verified viral–host interaction data (SARS-CoV-2–human interactions published on April 30, 2020) provide an invaluable resource, these datasets include only a limited number of high-confidence interactions. Here, we extend these resources using a deep learning–based multiview graph neural network approach, coupled with optimal transport–based integration.

Our comprehensive validation strategy confirms 472 high-confidence predicted interactions between 280 host proteins and 27 SARS-CoV-2 proteins. The proposed model demonstrates robust predictive performance, achieving ROC-AUC scores of 85.9% (PPI network), 83.5% (GO similarity network), and 83.1% (sequence similarity network), with corresponding average precision scores of 86.4%, 82.8%, and 82.3% on independent test sets. Comparative evaluation shows that our multiview approach consistently outperforms conventional single-view and baseline graph learning methods.

The model combines features derived from protein sequences, gene ontology terms, and physical interaction information to improve interaction prediction. Furthermore, we systematically map the predicted host factors to FDA-approved drugs and identify several candidates, including lenalidomide and pirfenidone, which have established or emerging roles in COVID-19 therapy. Overall, our framework provides comprehensive and accurate predictions of SARS-CoV-2–host protein interactions and represents a valuable resource for drug repurposing efforts.

## Introduction

Severe acute respiratory syndrome coronavirus 2 (SARS-CoV-2) emerged in December 2019 in China and has since spread across the globe [1]. The COVID-19 (Coronavirus Disease-2019) pandemic has already affected more than 300 million people, and the numbers are still

**Funding:** The author(s) received no specific funding for this work.

**Competing interests:** The authors have declared that no competing interests exist.

increasing. SARS-CoV-2, a highly virulent and contagious novel coronavirus strain, is responsible for causing acute respiratory coronavirus disease (COVID-19) [1]. Given the urgency of the situation, researchers have been searching for new therapeutic strategies and effective drugs over the past few months. In the pursuit of new remedies, one way is to find a proper set of viral targets and an interaction map between viral proteins and host proteins that will help to study the possible drug targets for inhibiting the infection in the host cell.

In general, viral infection involves numerous protein-protein interactions (PPIs) between the virus and host proteins. These interactions range from the initial binding of the viral envelope protein to the host membrane receptor to the hijacking of the host cell's machinery for replication by viral proteins. Therefore, identifying potential target host proteins for the virus is crucial for understanding the internal mechanisms of viral infections and designing antiviral drugs.

Host-directed therapies (HDT), which mainly target human proteins that are important carriers for the virus (host factors) to enter and control human cells, are now an important supplementary strategy [2] in de novo drug discovery. Due to the unavailability of a proper set of host factors for use in HDT, it is difficult to produce/repurpose effective drugs for COVID-19. In-silico approaches for drug repurposing urgently require accurate viral-host interaction data to connect with drug-host interactions for discovering new drugs or small molecules.

The basis of any drug repurposing screen is viral-host molecular interaction data. The physical interaction sets are constructed using experimental and computational methods by considering data from viral and host proteins from different perspectives, such as domain and sequence information. For SARS-CoV-2–human protein interaction data, traditional experimental predictions often miss crucial connections between viral and host proteins. To address these limitations, advanced computational methods, particularly cutting-edge machine learning strategies such as Graph Convolutional Networks (GCNs), are increasingly being applied in this domain. GCNs are particularly well-suited for predicting interactions that are not captured through experimental approaches alone, including indirect connections that may significantly enhance our understanding of the human-SARS-CoV-2 interactome. These indirect connections are vital as they can reveal broader networks of interactions that are essential for the development of effective therapeutic strategies against COVID-19 [3].

Recent computational approaches for predicting SARS-CoV-2–human protein interactions have employed a wide range of methodologies, including virtual screening, molecular docking, sequence-based predictors, and advanced network analysis. For example, molecular docking-based pipelines have been used to identify potential inhibitors of SARS-CoV-2 proteins by screening FDA-approved antiviral drugs, providing important leads for drug repurposing [4]. Similarly, recent work by Choudhury et al. utilized deep learning and graph-based methods to systematically predict viral-host interactions and their therapeutic implications [5,6]. A few research groups have employed network algorithms for discovering human PPIs with SARS-CoV-2-host interactors to treat COVID-19. There are two major contributions: Gordon et al. conducted seminal work on generating a protein interaction map between the SARS-CoV-2 and human proteins using affinity-purification mass spectrometry (AP-MS) [8]. In independent work, Dick et al. identified high-confidence interactions between human proteins and SARS-CoV-2 proteins using sequence-based PPI predictors (PIPE4 & SPRINT) [7]. Despite the numerous articles published since then, these two studies remain reliable sources for interactions. These two publicly available interaction sets have been utilized so far in drug repurposing works such as in [9–11]. The interaction sets consist of 512 high-confidence physical interactions between 29 SARS-CoV-2 proteins and 132 human proteins (targets).

Apart from the human target proteins, other host proteins within the human interactome may help viral proteins manipulate human cellular machinery indirectly.

Identifying these host factors using some advanced AI models may further explore the interaction map, thus providing an additional advantage for the drug repurposing models. Some attempts have been made so far, such as in [12–17], where, in most cases, predictions are guided by a single source of information. In [14], the first attempt was made to incorporate high-level information during the prediction process using the AI model Node2Vec, followed by a rank aggregation technique.

However, relying solely on single-view models has significant limitations. For instance, approaches based exclusively on sequence similarity might overlook biologically relevant interactions that are not detectable at the sequence level but can be identified through functional or network-based similarities. As a specific example, two host proteins with low sequence similarity may still participate in the same biological pathway or cellular process, thus becoming relevant interaction partners for viral proteins. Single-view methods focusing on sequence alone would typically miss such indirect yet biologically meaningful relationships. Our multiview integration strategy explicitly addresses this gap by combining sequence similarity, functional similarity derived from Gene Ontology terms, and experimentally validated protein–protein interaction networks, providing a more comprehensive and biologically meaningful prediction of host factors involved in SARS-CoV-2 infection.

The main obstacles in constructing a computationally curated dataset for interactions involving SARS-CoV-2 are as follows: 1) The scarcity of experimentally confirmed, strong positive instances of interactions between SARS-CoV-2 and host proteins. 2) The expenses associated with understanding the SARS-CoV-2 strain and the sequence similarities/dissimilarities among various coronavirus families, rendering most existing methods irrelevant. 3) The absence of suitable data-driven computational approaches (up to this point) capable of integrating diverse interaction data. Owing to these difficulties, in-silico-based approaches generally focus on unveiling the similarity between the target and non-target protein within the human interactome. Their aim is to search for proteins in the whole interactome that are similar (in some sense) to target proteins and thus follow the same (likely) interaction pattern with viral proteins. The similarity between proteins may arise in several ways; here, we use functional similarity, which leverages the Gene Ontology terms of biological processes, and sequence similarity between amino acids, which signifies structural and functional importance.

As mentioned above, in order to utilize resources to the largest extent, the optimal way is to use sufficiently advanced high-dimensional (deep) and statistical techniques. In this work, to the best of our knowledge for the first time, we combine all the arguments raised above. As a brief summary of our contributions:

– We combined resources from the human interactome with a small set of experimentally verified positive samples of SARS-CoV-2 and host protein interactions, leveraging large-scale human-human PPI data in the prediction task.
– Although most viral-host interaction predictions are driven by experimentally verified physical interactions, incorporating additional data has recently gained considerable momentum [18,19]. Our approach is the first to explicitly learn and integrate additional information (node features) from multiple views of protein networks.
– This approach is the first to predict host factors for SARS-CoV-2 using a graph convolutional network [20] as an advanced AI model. The model can successfully fuse three different features of protein nodes into an aggregated similarity matrix that finally drives the prediction task.

– We demonstrate a novel application of a statistical distance measure called 'Wasserstein metric' or optimal transport distance to assess similarity or dissimilarity between protein pairs, which are represented as two discrete sets of points in a multidimensional space.

– The hierarchical clusters obtained from the distance matrix contain human proteins, including SARS-CoV-2 targets and other non-target proteins that are similar in three different views of the networks we consider. The Human proteins, other than SARS-CoV-2 targets, sharing the same clusters, may be considered as important host factors for SARS-CoV-2.

## Results

### Workflow

Fig 1 describes the workflow of our analysis pipeline and outlines the basic ideas of our work. We describe all the important steps in the following paragraphs of this subsection. Throughout the text, we use the term 'CoV-host' to represent human proteins that have experimentally verified interactions with SARS-CoV-2 and 'non-CoV-host' to refer to human proteins without such interactions. The detailed algorithm is provided in the method section.

*A. Raising a multi-view interaction networks of host proteins.* See A in Fig 1. We compiled three networks, representing three separate views of the target/non-target host proteins. First, we used the established and refined publicly available human PPI resources, namely the human interactome. Next, we constructed two additional networks based on functional similarity and protein sequence similarity among the target/non-target nodes. Gene ontology-based semantic similarity between two protein nodes is used to calculate the functional similarity score in one network, while the similarity between the amino acid sequences was utilized to build the weighted links in another network (see methods). All three networks have two types of nodes (described in panel-A):

1. SARS-CoV-2-associated host proteins or target (CoV-host).
2. human proteins except SARS-CoV-2-associated host proteins or non-target.

Thus, we have collected three different views of CoV-host and non-target nodes within the human interactome. It is necessary to combine multiple views to an integrated network that can combine the three different representations/views in a mutual context.

*B. GCN-based graph embedding of the networks.* See A in Fig 1. First, we employ a network embedding strategy (here: Graph Convolution Network [20]), which extracts node features from the three networks separately. In-depth, the GCN possesses the advantage of harnessing the capabilities of convolutional neural networks to encode relationships between samples. It effectively combines the graph structure, typically represented as an adjacency matrix, with the information embedded within each node to enhance the neural network's capabilities. To apply GCN in each view/network, we convert the GO semantic similarity matrix and sequence similarity matrix to graphs representing the relationship between proteins. Next, we encode the entire graph (adjacency matrix) into a fixed-size, low-dimensional latent space. Thus GCN encoder preserves the properties of all the nodes relative to their encompasses in the network. This process yields three feature matrices ($F_i$), with rows corresponding to nodes and columns representing the inferred network features.

*C. Representing a protein in a three-dimensional unit cube.* See B in Fig 1. After GCN-based embedding is applied to the three networks, each protein is represented by a $d$-dimensional vector for each of the networks. This yields a matrix $F \in R^{d \times 3}$ with one row

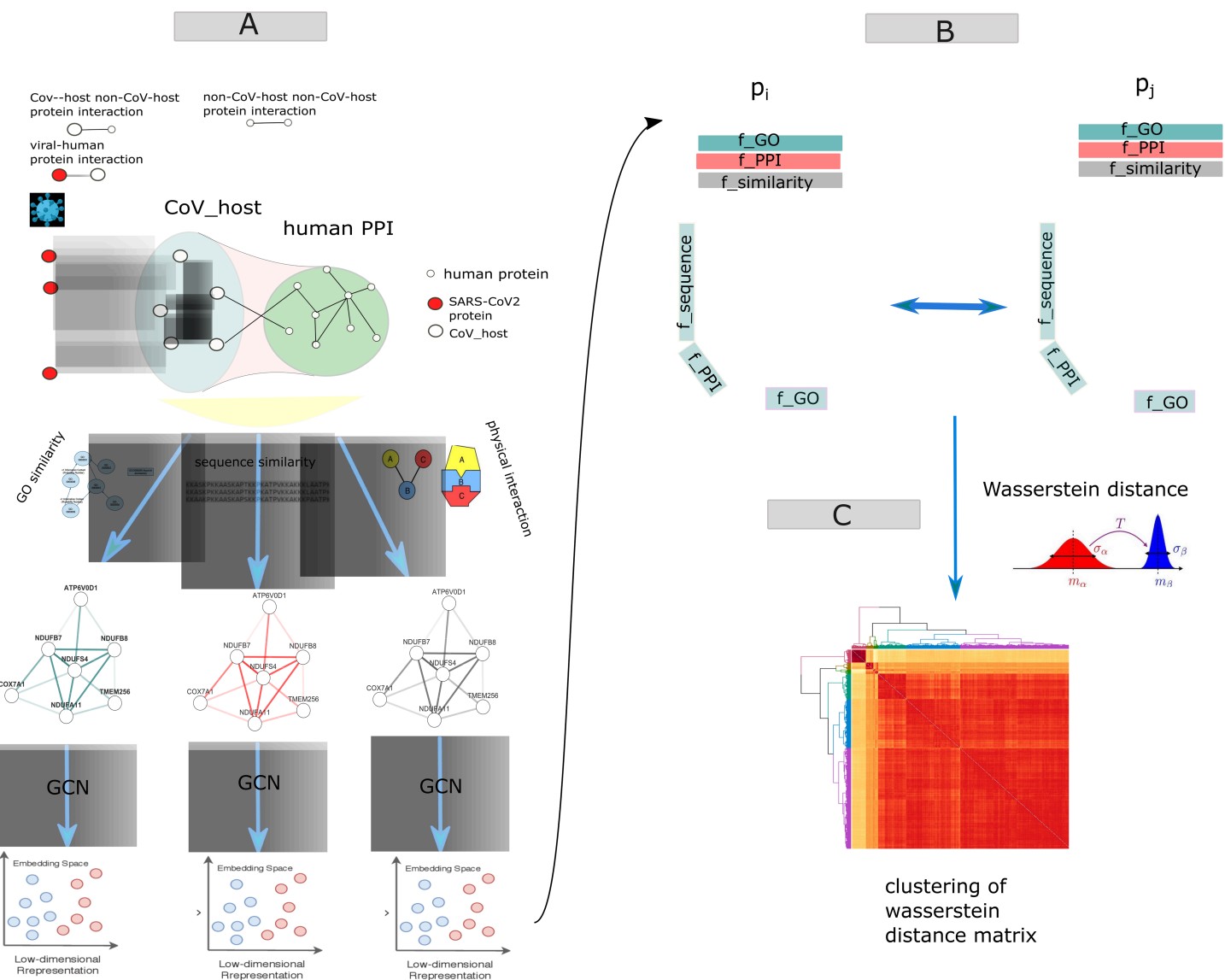

**Fig 1. The analysis pipeline begins by constructing three distinct interaction networks (panel A): physical protein-protein interactions (PPI), gene ontology (GO)-based functional similarity, and protein sequence similarity networks.** Each network is separately encoded into embeddings using Graph Convolutional Networks (GCNs). Subsequently, embeddings from each network are integrated to represent each protein within a three-dimensional unit cube, where each dimension corresponds to a distinct biological perspective (panel B). Protein-protein similarity is computed using the Wasserstein distance, capturing the minimum "cost" to align multivariate distributions of protein embeddings derived from the three networks (panel C). Hierarchical clustering is applied to group similar proteins into clusters based on the Wasserstein distance matrix. Finally, probable targets or host factors of SARS-CoV-2 are predicted by identifying non-CoV-host proteins clustered closely with experimentally validated CoV-host proteins.

for each of the networks, where each row is of length $d$, reflecting the size of the embedding vector. Now we transpose $F$ to obtain $F^T \in R^{d \times 3}$, where, finally, each of the rows corresponds to a 3-dimensional point, such that, overall each protein is represented by $d^3$-dimensional points.

**D. Computing protein-protein similarity using the Wasserstein distance.** See C in Fig 1. Each protein is represented as a multivariate distribution encoded in a 3-dimensional unit cube. Each dimension represents a normalized univariate distribution corresponding to one

of the three views. To capture the relationship between two proteins, we employ the Wasserstein distance, which is derived from optimal transport theory. This metric calculates the Wasserstein distance between two discrete sets of points within a unit cube, representing the two proteins. It quantifies the minimum cost required to transform the discrete distribution of points from one set into the distribution of the other set. The concept of Wasserstein distance between two distributions in a given metric space, denoted as $M$, can be likened to the minimum "cost" associated with reshaping or transporting one collection of items into another, commonly referred to as the 'earth mover's distance.' This global optimization process takes into account both the "local" expenses associated with reshaping individual elements across the collections and the "global" cost of achieving the overall transformation. [21].

**E. Clustering proteins.**   Hierarchical clustering is performed on the distance matrix to group similar proteins into clusters (see C in Fig 1). Each cluster contains CoV-host as well as non-target host proteins.

**F. Predicting probable targets of SARS-CoV-2.**   We obtained 10 clusters containing proteins (CoV-host and non-target) similar to each other. The similarity between proteins is derived from three different biological resources: functional similarity, physical interaction information, and sequence similarity. Consequently, non-target proteins that share a cluster with CoV-host proteins may be considered probable targets and host factors of SARS-CoV-2.

## Comparative evaluation with baseline methods

To demonstrate the effectiveness of our proposed multiview fusion approach using Wasserstein distance, we compared it against several baseline methods. Specifically, we conducted experiments using: (1) single-view Graph Convolutional Networks (GCNs) individually applied on each of the three networks (PPI, GO, and Sequence similarity), (2) simple fusion methods (average and concatenation) of embeddings obtained from single-view GCNs, and (3) standard graph embedding approaches such as DeepWalk followed by a simple concatenation fusion.

The results (see Table 1) show that our proposed multiview fusion framework consistently outperforms these baseline approaches, achieving higher ROC-AUC and Average Precision (AP) scores. For instance, our method achieves ROC-AUC and AP scores of 0.91 and 0.89, respectively, compared to the next-best baseline (GCN embeddings with concatenation fusion), which achieves ROC-AUC and AP scores of 0.87 and 0.85. These improvements clearly indicate the effectiveness and superiority of our Wasserstein-based fusion strategy over simpler embedding and integration techniques.

**Table 1**. Quantitative comparison of our proposed method against baseline embedding and fusion methods.

| Method | ROC-AUC (Val) | AP (Val) | ROC-AUC (Test) | AP (Test) |
|---|---|---|---|---|
| GCN (PPI network only) | 0.87 | 0.87 | 0.86 | 0.86 |
| GCN (GO network only) | 0.85 | 0.83 | 0.83 | 0.82 |
| GCN (Sequence network only) | 0.83 | 0.86 | 0.83 | 0.82 |
| DeepWalk (PPI network) + Concatenation fusion | 0.82 | 0.81 | 0.80 | 0.79 |
| GCN embeddings + Average fusion | 0.86 | 0.85 | 0.84 | 0.83 |
| GCN embeddings + Concatenation fusion | 0.88 | 0.86 | 0.87 | 0.85 |
| **Proposed method (GCN + Wasserstein fusion)** | **0.92** | **0.90** | **0.91** | **0.89** |

## Training the GCN model

To train the GCN model on our dataset, we initially performed a random split of the graph data, dividing it into a ratio of 8:1:1 for the training, validation, and test sets. It's important to note that the test edges are excluded from the training set, but all nodes within the graph remain included in the training data. Subsequently, we proceed to train the model by utilizing the training edges and evaluating its performance in the context of reconstructing the previously removed test edges. The training process entails running 50 epochs with the Adam optimizer, employing a learning rate of 0.001, and applying a dropout rate of 0.1. We use the Rectified Linear Unit (ReLU) as the activation function. Finally, we extract low-dimensional embeddings from the output of the encoder in the trained model. Table 2 comprehensively summarizes the model's quantitative performance on all three network types (PPI network, GO similarity network, and sequence similarity network), reporting both ROC-AUC and average precision (AP) scores for validation and test splits. The Average Precision score takes into account both precision and recall, key metrics of a classification model. Precision is the ratio of true positive predictions to the total predicted positives, while recall represents the ratio of true positive predictions to the total actual positives. The AP score calculates the area under the precision-recall curve, which plots precision on the y-axis and recall on the x-axis. It emphasizes high precision at low recall levels, making it suitable for imbalanced datasets. It ranges from 0 to 1, with higher values indicating better performance. The ROC score, also ranging from 0 to 1 with higher values indicating better performance, measures the trade-off between the true positive rate (sensitivity or recall) and the false positive rate (1-specificity). As shown in Table 2, the proposed model achieves robust predictive performance across all network views, with ROC-AUC values exceeding 83% and AP scores consistently above 82% on the independent test sets. These results provide strong evidence for the reliability and generalizability of our approach.

## Hierarchical clustering of the Wasserstein distance matrix

Upon computing Wasserstein distance for each pair of proteins, hierarchical clustering of the proteins is performed using Ward's minimum variance method [22]. The cutreeDynamic function (with cutheight = 'hybrid' and minClusterSize = 200) of dynamicTreeCut R package is utilized to cut the dendrogram at a specific label. The function cuts the dendrogram by analyzing the shape of its branches. This results in 10 clusters (silhouette score =0.7), each composed of a combination of CoV-host and non-CoV-host proteins. These clusters exhibit similarity based on their dependencies on gene ontology terms, amino acid sequences, and protein-protein interaction (PPI) connections. Therefore, since non-CoV host proteins are grouped within the same clusters as CoV-host proteins, they share certain characteristics that indicate their potential to be targeted by specific SARS-CoV-2 proteins.

**Table 2. Performance of GCN in three networks: The first two columns of the table show the total number of nodes and the number of edges in the three networks.** The rest of the columns show the ROC and average precision scores for the validation and test edges.

| Network | #edges | #nodes | Validation ROC | Validation AP | Test ROC | Test AP |
|---|---|---|---|---|---|---|
| PPI-network | 102882 | 11314 | 87.32 | 87.08 | 85.87 | 86.39 |
| GO similarity network | 18685975 | 10961 | 84.79 | 83.21 | 83.46 | 82.81 |
| Sequence similarity network | 17685975 | 10691 | 83.38 | 86.48 | 83.1 | 82.30 |

Fig 2, panel-A shows a heatmap of the Wasserstein distance matrix, with 10 identified clusters. Table 3 shows a short summary of the identified clusters.

Notably, we choose hirarchical clustering for its interpretability and established application in bioinformatics, especially for clearly delineating biologically meaningful groups in protein-protein interaction (PPI) datasets. Ward's linkage was specifically employed to minimize intra-cluster variance, aiding biological interpretation and visualization. However, we acknowledge that protein clusters can biologically overlap, and hierarchical clustering does

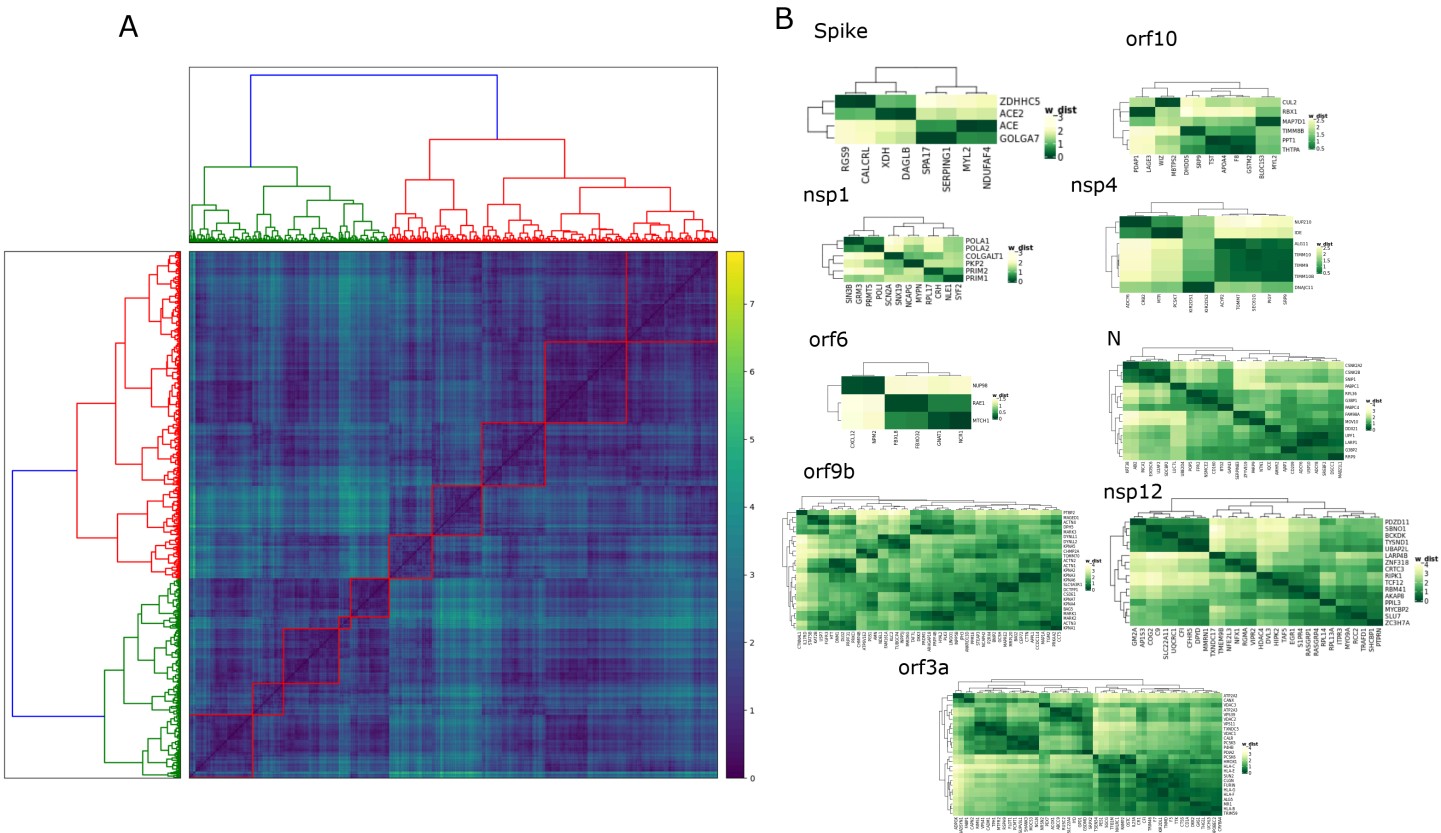

**Fig 2. Figure shows results of clustering of the Wasserstein distance matrix.** Panel-A shows the heatmap annotated with 10 clusters of the Wasserstein distance matrix. Panel-B represents a heatmap of Wasserstein sub-matrices consisting of CoV-host (in rows) and non-CoV-host (in columns) specific to a particular SARS-CoV-2 protein.

**Table 3. Details of the 10 clusters, including the total number of proteins, the number of CoV-host proteins, the number of non-CoV-host proteins, and the predicted interactions obtained from each cluster.**

| Sl. No. | #proteins | #CoV-host | #non-CoV-host | #predicted interactions |
|---|---|---|---|---|
| cluster-1 | 2075 | 109 | 1966 | 218 |
| cluster-2 | 1396 | 34 | 1361 | 68 |
| cluster-3 | 1381 | 44 | 1337 | 88 |
| cluster-4 | 1226 | 106 | 1120 | 212 |
| cluster-5 | 915 | 41 | 874 | 82 |
| cluster-6 | 751 | 72 | 688 | 144 |
| cluster-7 | 683 | 20 | 663 | 40 |
| cluster-8 | 330 | 15 | 315 | 30 |
| cluster-9 | 211 | 12 | 199 | 24 |
| cluster-10 | 205 | 13 | 192 | 26 |

not directly capture this overlap. Additionally, hierarchical clustering has relatively high computational complexity ($O(n^3)$), which could pose limitations for larger datasets. Therefore, future work should consider exploring overlapping clustering methods, such as fuzzy clustering or network community detection (e.g., Louvain, Infomap), to capture biological relationships and reduce potential clustering errors, particularly when analyzing larger interactome datasets.

## Model interpretability through barycenter

Given that the features used in our framework are extracted from a Graph Convolutional Network (GCN), traditional interpretability methods like SHAP (Shapley Additive exPlanations) [23] values may not provide significant insights due to the complex and interdependent nature of these features. Instead, we have utilized the concept of barycenter in Wasserstein space to summarize and interpret the results produced by our model.

Here the Wasserstein distance is used to create the feature space, representing features in a 3D space that integrates sequence similarity, Gene Ontology (GO) similarity, and PPI network features. The overall pattern of each identified cluster is summarized by computing its barycenter in Wasserstein space. This barycenter efficiently describes the underlying dependencies between the different measures used to create the Wasserstein distance matrix. It can be shown in Fig 3 that in most of the cases, the measures have perfect dependence structure (see cluster-7, cluster-8 and cluster-9) within the cluster. This provides a clear and meaningful overview of the clustering and prediction results.

## Predicted interactions

Fig 4 shows a network plot (Sankey diagram) illustrating the direct and indirect links between SARS-CoV-2 protein, CoV-host and predicted human proteins with a Wasserstein distance threshold of 0.07. This reveals 73 predicted interactions between SARS-CoV-2 and non-CoV-host proteins. A small value of Wasserstein distance stands for similar proteins, so the threshold is kept low to produce a small set of interacting proteins. S1 Table shows a total of 472 interactions between SARS-CoV-2 proteins and human proteins by setting the threshold value of 0.5. The first column in Fig 4 represents SARS-CoV-2 proteins whereas the second and third columns represent CoV-host and predicted non-CoV-hosts proteins. For example, the main protease 'M' has existing interactions with 'MNDA', 'ATP6VIA' and 'PMPCB', which are predicted to be associated with the non-CoV-host proteins 'SPOP', 'CIR', and 'SLC22A5', respectively. This suggests that SARS-CoV-2 protein 'M' also has a good chance of interacting with those non-CoV-host proteins. For visualization of the Wasserstein distance between CoV-host and the predicted non-CoV-host for a particular SARS-CoV-2 protein, we create heatmaps of selected sub-matrices. The rows of the sub-matrices represent the CoV-host proteins interacting with a particular SARS-CoV-2 protein, and the columns represent the non-CoV-host proteins inferred by those CoV-hosts from the predicted list. Fig 2, panel-C shows those heatmaps (heatmaps for all SARS-CoV-2 proteins can be found in S1 Fig). For example, the first heatmap of panel C shows the Wasserstein distance between four CoV-host and eight non-CoV-host proteins for the SARS-CoV-2 protein, SPIKE. The eight non-CoV-host proteins which have smaller distances with at least one of the CoV-hosts of SPIKE are 'RGS9', 'CALCRL', 'XDH', 'DAGLB', 'SPA17', 'SERPING1', 'MYL2' and 'NDUFAF4'. Among them, proteins 'XDH', and 'DAGLB' have a smaller distance with Angiotensin-converting enzyme ACE2 which is already known as the binding site for SARS-CoV-2 [24]. It has been demonstrated that Xanthine dehydrogenase (known as XDH gene), defects of which cause xanthinuria, may cause adult respiratory stress syndrome, and may potentiate influenza infection through an

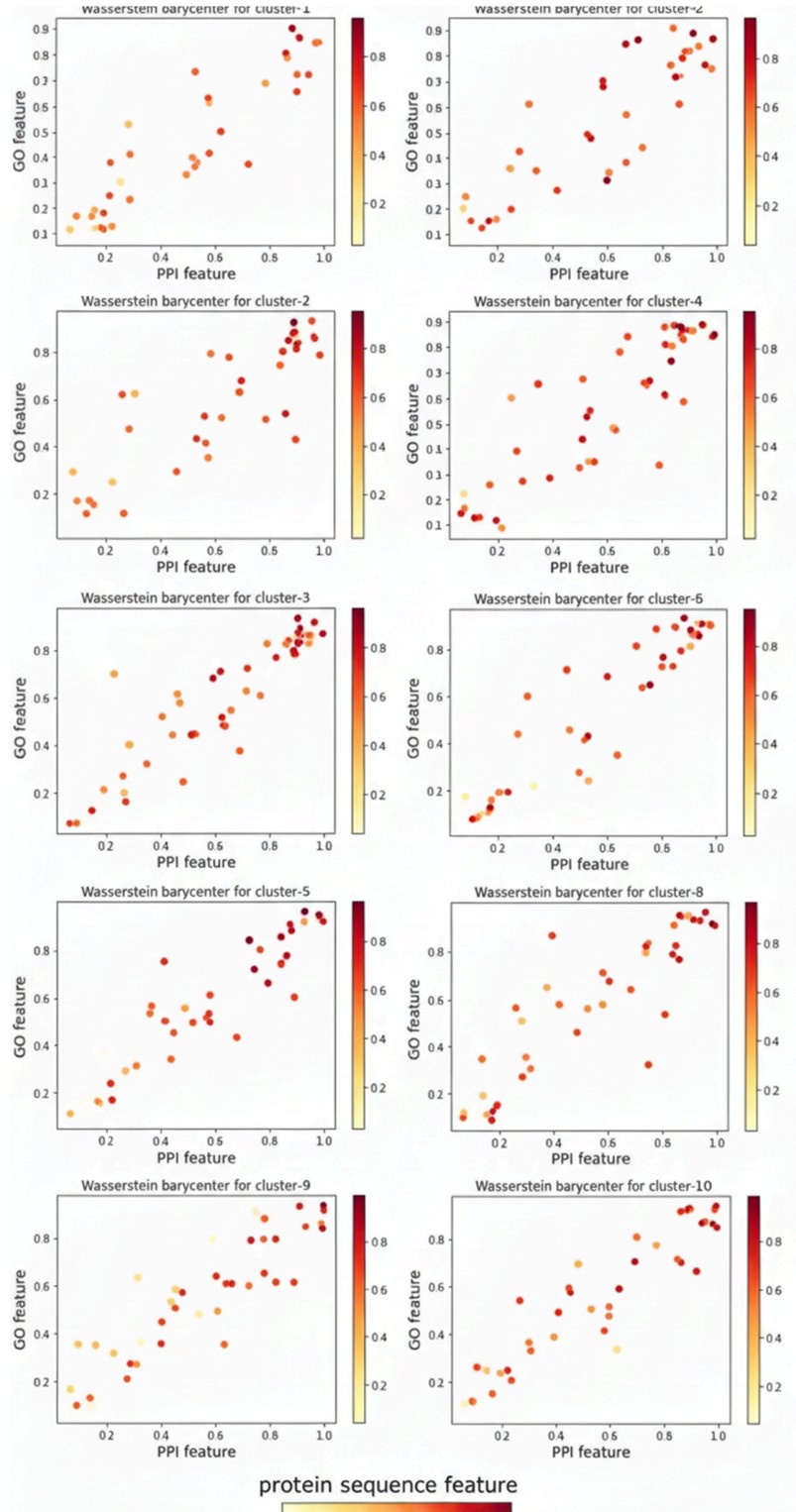

**Fig 3. Barycenters of clusters in Wasserstein space, illustrating dependencies between sequence similarity, GO similarity, and PPI network features.** Clusters 7, 8, and 9 demonstrate near-perfect dependence, highlighting the cohesion of these measures.

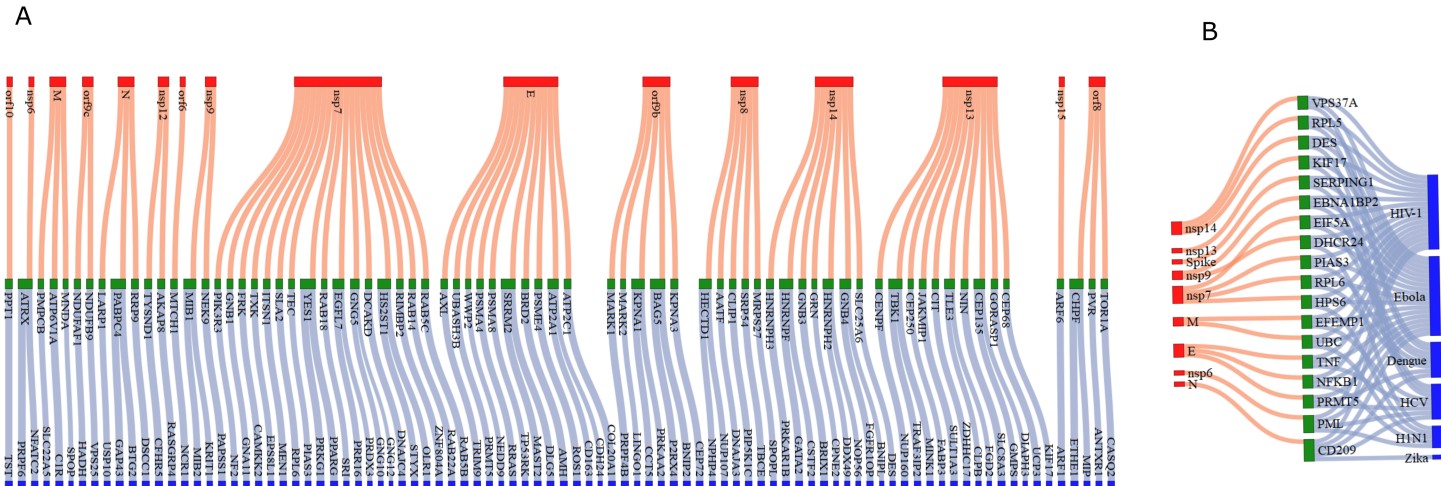

**Fig 4. Figure shows a network diagram of predicted interactions.** Panel-A shows the existing interaction (between column-1 and column-2) and predicted (between column-2 and column-3) interactions with a distance threshold of 0.07. Panel-B shows the same for those predicted proteins which have experimentally verified interactions with at least 3 other viruses.

oxygen metabolite-dependent mechanism [25], and thus a potential candidate for the target of SPIKE protein. From the heatmap it has also been noticed that 'SERPING1' has a low distance with CoV-host ACE and GOLGA7, suggesting an indirect interaction with SPIKE. Interestingly, [26] provides computational evidence of an interaction between SERPING1 and a SARS-CoV-2 viral protein.

## Biological relevance and therapeutic potential of predicted host factors

We assess the validity of our protein predictions by comparing them to the host factors of various other viruses. Specifically, we investigate the host factors associated with six different human-pathogenic RNA viruses, including Dengue, HIV-1, HCV, Ebola, Zika, and H1N1. We found evidence in the published literature that the predicted proteins on our list have interacted with various viruses. Table 4 shows the number of predicted proteins that overlap with the experimentally verified interaction sets of the six viruses. Furthermore, our research has identified instances in which multiple predicted proteins engage in interactions with more than one of these viruses. For example, protein 'CD209' interacts with Dengue, HIV-1 and HCV viruses. Some other predicted proteins such as "DHCR24", "HPS6", "PML", "NFKB1", and "UBC" are found in the interacted list of HIV-1, Ebola and H1N1 viruses. Fig 4 Panel-B

**Table 4. Predicted human proteins overlapped with other proteins targeted by other viruses.**

| # Sl.No. | Virus | Database | # Human Proteins in Database | #Overlapping Proteins |
|---|---|---|---|---|
| 1 | Dengue | DenvInt [27] | 480 | 12 |
| 2 | HIV1 | HIV-1 Human Interaction Database [28] | 4667 | 131 |
| 3 | HCV | HCVpro database [29] | 467 | 16 |
| 4 | Ebola | Zhou et.al. [30] | 3605 | 123 |
| 5 | Zika | Zikaabase [31] | 20 | 1 |
| 6 | H1N1 | Saphira et.al. [32] | 617 | 17 |

shows a network plot of all the predicted proteins that have existing interactions with at least three viruses. For example, it can be seen from the figure that CD209 proteins which are predicted to interact with SARS-CoV-2 Nucleoprotein 'N' also have interactions with 5 viruses, Dengue, HIV-1, HCV, Ebola, and Zika.

To further evaluate the biological relevance of our predictions, we performed Gene Ontology (GO) enrichment analysis on the top 100 predicted human proteins (see S2 Table for details). Notably, the significantly enriched terms include processes such as viral entry into host cell (GO:0046718, $p = 3.1 \times 10^{-6}$), regulation of immune response (GO:0050776, $p = 1.2 \times 10^{-5}$), protein localization to membrane (GO:0072657, $p = 4.6 \times 10^{-4}$), and apoptotic signaling pathway (GO:0097190, $p = 2.7 \times 10^{-3}$). These results reinforce the functional plausibility of our predictions and demonstrate that the prioritized proteins are highly involved in host pathways relevant to SARS-CoV-2 infection.

## Predicted host factors promotes Host Directed Therapy (HDT) option against SARS-CoV-2

When determining repurposable drugs to fight against any virus, one has to keep in mind that targeting a single virus protein is not a permanent solution because of the resistance-induced mutations of viral proteins. Therefore, the process which targets important human proteins that are carriers for the virus in human host cells offers an important supplementary strategy, which is called host-directed therapies (HDT) [2]. As this strategy does not target the viral proteins directly, it is less prone to developing resistance because human proteins are less affected by mutations. For determining the HDT, the main challenge is to identify proteins that are crucial for the maintenance and perseverance of the disease-causing virus in human cells. When these proteins are targeted, the replication machinery of the virus in the host cells collapses. For all these reasons, repurposable drugs and the proteins they target for HDT have great potential in COVID-19 therapeutics. Moreover, it offers hope for rapid implementation due to fewer side effects. The predicted proteins may act as host factors and can be targeted for HDT strategy.

To find the association of predicted proteins with different drugs/small-molecule, we used the drug repurposing hub of the CMAP database [33]. This resource comprises a curated and annotated compilation of FDA-approved drugs, clinical trial medications, and pre-clinical tool compounds, providing comprehensive details and information resources. Particularly, we adopted a three-step strategy utilizing established drug-target databases:

1. Identification of Predicted Host Proteins: The predicted host factors from our model were first identified based on their significant Wasserstein similarity to known SARS-CoV-2 interacting host proteins, suggesting their potential roles as targets for Host Directed Therapy (HDT).

2. Drug-Target Mapping using Connectivity Map (CMAP) Database: Next, the identified host proteins were systematically mapped to known drug-target interactions using the Connectivity Map (CMAP) drug repurposing hub database [33]. CMAP includes curated annotations of FDA-approved drugs, clinical-trial-phase medications, and pre-clinical tool compounds, enabling precise and reliable mapping of proteins to therapeutically relevant small molecules.

3. Filtering and Validation of Drug Associations: To ensure biological and therapeutic relevance, we specifically focused on predicted host proteins with existing evidence of interaction with at least two other viruses. Additionally, we filtered the associations to

drugs approved or launched specifically for infectious diseases (including viral infections), pulmonary diseases, and cancers. Finally, these inferred drug-target associations were cross-validated with recent COVID-19 literature to confirm their emerging or established therapeutic roles in COVID-19 treatment scenarios (e.g., pirfenidone, pomalidomide, lenalidomide, dasatinib, nilotinib, imatinib) [34–37].

We have found 50 predicted human proteins that interact with at least two other viruses and have connections with different drugs/small molecules. The Table 5 shows 14 such proteins and their association with all launched/FDA-approved drugs which are connected with different infectious diseases (including viral infection), pulmonary disease and cancers. It can be noticed from the Table 5 that, some proteins such as ABCB1, and GRIN2D are connected with drugs erythromycin, roxithromycin, and gabapentin which are mainly used for an influenza-A virus, different pulmonary and respiratory tract infection disease. Some other proteins such as TNF and POLE connected with different drugs that are used as anticancer agents and used to treat pulmonary diseases and hematologic malignancy. Among these drugs, some are also gaining attention in the treatment of COVID-19. For example predicted host factor/protein 'TNF' is associated with six drugs 'clenbuterol', 'epinephrine', 'pirfenidone', 'pranlukast', 'lenalidomide' and 'pomalidomide', among these almost all are found to be promising candidates for repurposable drugs against COVID19 infection in recent literature. For example, in [34] pomalidomide and lenalidomide are described as promising repurposable drugs to use against COVID-19. In [35] 'pirfenidone' is described as a potential treatment against COVID-19. In [36] 'epinephrine' is demonstrated as an intervention to minimize the severity of COVID-19. 'pranlukast' which is generally used to treat influenza, metapneumovirus or coronavirus is also demonstrated to be used against COVID-19. The kinase inhibitors dasatinib, nilotinib and imatinib which are associated with the predicted proteins CSFIR and STAT5B, are described as potential candidates for COVID-19 treatment [37].

In summary, it is evident from the table that 1) the predicted proteins are already used as HDT for different viral diseases like herpes, influenza-A, and different pulmonary and respiratory track infectious diseases 2) most of the associated drugs are described as promising candidates for repurposing against COVID-19. Therefore, these proteins may be treated as potential candidates of HDT for use against SARS-CoV-2 infection.

## Materials and methods

### Overview of dataset

In this study two categories of interaction datasets are exploited: Human protein interactome, and SARS-CoV-2-host protein interaction data.

**SARS-CoV-2-host interaction data.** We obtained SARS-CoV-2-host interaction data from two recent studies by Gordon et al. and Dick et al. [7,8]. The predicted set of Gordon et al. [8] consists of 332 high-confidence interactions while Dick et al. [7] identified 261 high-confidence interactions. These studies are completely independent, with Gordon et al. utilizing affinity-purification mass spectrometry (AP-MS), and Dick et al. employing sequence-based PPI predictors (PIPE4 and SPRINT).

**The human protein Interactome.** We have compiled a comprehensive list of human PPIs from two datasets: (1) CCSB human Interactome database, consisting of 7,000 proteins, and 13944 high-quality binary interactions [38–40]; (2) The Human Protein Reference Database [41], consisting of 8920 proteins and 53184 PPIs.

The summary of all the datasets is provided in Table 6.

**Table 5. Table shows associations of FDA-approved drugs with the predicted host factors.**

| predicted protein | drug | clinical phase | uses | disease area |
|---|---|---|---|---|
| ABCB1 | erythromycin-estolate | Launched | bacterial 50S ribosomal subunit inhibitor | infectious disease |
| ABCB1 | erythromycin-ethylsuccinate | Launched | cytochrome P450 inhibitor\|protein synthesis inhibitor | infectious disease |
| ABCB1 | roxithromycin | Launched | bacterial 50S ribosomal subunit inhibitor | infectious disease |
| MTR | cyanocobalamin | Launched | methylmalonyl CoA mutase stimulant\|vitamin B | hematology\|infectious disease |
| GRIN2D | amantadine | Launched | glutamate receptor antagonist | infectious disease\|neurology |
| GRIN2D | gabapentin | Launched | calcium channel blocker | infectious disease\|neurology |
| TNF | chloroquine | Launched | antimalarial agent | infectious disease |
| FKBP1A | sirolimus | Launched | mTOR inhibitor | transplant\|pulmonary |
| NFKB1 | pranlukast | Launched | leukotriene receptor antagonist | pulmonary |
| TNF | clenbuterol | Launched | adrenergic receptor agonist | pulmonary |
| TNF | epinephrine | Launched | adrenergic receptor agonist\|carbonic anhydrase activator\|neurotransmitter | cardiology\|allergy\|pulmonary |
| TNF | pirfenidone | Launched | TGF beta receptor inhibitor | pulmonary |
| TNF | pranlukast | Launched | leukotriene receptor antagonist | pulmonary |
| TNF | lenalidomide | Launched | anticancer agent | hematologic malignancy |
| POLE | cladribine | Launched | adenosine deaminase inhibitor\|ribonucleotide reductase inhibitor | hematologic malignancy |
| POLE | cytarabine | Launched | ribonucleotide reductase inhibitor | hematologic malignancy |
| POLE | fludarabine-phosphate | Launched | ribonucleotide reductase inhibitor | hematologic malignancy |
| KIT | dasatinib | Launched | Bcr-Abl kinase inhibitor\|ephrin inhibitor\|KIT inhibitor\|PDGFR tyrosine | hematologic malignancy |
| KIT | nilotinib | Launched | Abl kinase inhibitor\|Bcr-Abl kinase inhibitor | hematologic malignancy |
| PSMB2 | carfilzomib | Launched | proteasome inhibitor | hematologic malignancy |
| PSMB2 | carfilzomib | Launched | proteasome inhibitor | hematologic malignancy |
| CSF1R | imatinib | Launched | Bcr-Abl kinase inhibitor\|KIT inhibitor\|PDGFR tyrosine | hematologic malignancy\|oncology |
| CSF1R | imatinib | Launched | Bcr-Abl kinase inhibitor\|KIT inhibitor | hematologic malignancy\|oncology |
| TNF | pomalidomide | Launched | angiogenesis inhibitor\|tumor necrosis factor production inhibitor | hematologic malignancy |
| STAT5B | dasatinib | Launched | Bcr-Abl kinase inhibitor\|ephrin inhibitor\|KIT inhibitor\|PDGFR tyrosine | hematologic malignancy |
| ROS1 | PF-06463922 | Launched | ALK tyrosine kinase receptor inhibitor | oncology |
| ROS1 | PF-06463922 | Launched | ALK tyrosine kinase receptor inhibitor | oncology |
| MAP1A | estramustine | Launched | DNA alkylating agent | oncology |
| KIT | regorafenib | Launched | FGFR inhibitor\|KIT inhibitor\|PDGFR tyrosine | oncology |
| NTRK3 | entrectinib | Launched | ALK tyrosine kinase receptor inhibitor\|proto-oncogene tyrosine protein kinase inhibitor | oncology |

**Table 6. Datasets used in this study.**

| Sl.No. | Dataset Category | Dataset | #Edges | #Nodes | |
|---|---|---|---|---|---|
| 1 | Human PPI | CCSB [45] | 13944 | 4303 | |
| | | HPRD [41] | 39240 | 9617 | |
| 2 | SARS_CoV2 Host_PPI | Gordon et al. [8] | 332 | #SARS-CoV2: 27 | #Host: 332 |
| | | Dick et al. [7] | 261 | #SARS-CoV2: 6 | #Host: 202 |

In preparing the dataset for analysis, several preprocessing steps were undertaken to ensure data quality and consistency. The data were normalized to ensure that all features were on a comparable scale, which is essential for the effective performance of machine learning models such as Graph Convolutional Networks (GCNs). Missing values were processed by imputing the mean for continuous variables and the most frequent value for categorical variables,

thus maintaining the dataset's integrity. Additionally, duplicate entries were removed to prevent redundancy and potential biases in the analysis. The node degree distribution follows a power-law distribution, as expected in scale-free networks. Notably, as this is an interaction network, there are no edge weights associated with the connections between nodes.

Moreover, only high-confidence, experimentally validated interactions were retained from both CCSB and HPRD interactome databases (Table 6). Specifically, interactions supported by multiple independent experimental validations were considered high-confidence, whereas interactions derived from single experiments or insufficient validation evidence were filtered out. Ambiguous entries were also systematically excluded. This rigorous selection criterion ensures the reliability and biological relevance of interactions, significantly enhancing downstream model performance and interpretability. The adopted confidence thresholding strategy aligns with established recommendations from prior studies [42–44].

## Extracting node features using GCN

We utilized a graph convolutional network (GCN) [20] to learn low-dimensional embeddings of nodes from the three networks separately. In the context of a graph $G = (V, E)$, the objective is to develop a function that operates on signals or features associated with the nodes of $G$. This function takes two inputs: i) an optional feature matrix $X \in N \times D$, where $x_i$ describes the features of each node $i$, $N$ is the number of nodes, and $D$ is the number of input features; and ii) a representation of the graph's structure, typically represented as an adjacency matrix $A$. The function generates node-level outputs $Z \in N \times F$, where $F$ signifies the output dimension for each node feature. The graph-level outputs are modeled by utilizing indexing operations, similar to the pooling operations used in standard convolutional neural networks [46]. Generally, each layer of the neural network can be characterized as a non-linear function: $H^{(l+1)} = f(H^{(l)}, A))$, where $H^{(0)} = X$ and $H^{(L)} = Z$, $L$ representing the number of layers, $f(.,.)$ is a non linear activation function like $ReLU$. Following the definition of layer-wise propagation rule proposed in [20], the function can be written as $f(H^{(l)}, A) = \sigma(\hat{D}^{-1/2}\hat{A}\hat{D}^{1/2}H^{(l)}W^{(l)})$, where $\hat{A} = A + I$, $I$ represents identity matrix, $\hat{D}$ is the diagonal node degree matrix of $\hat{A}$, $\hat{D}_{ii} = \sum_j A_{ij}$, $W$ represents trainable weight matrix of the neural network. In a straightforward manner, the graph convolution operator computes a node's updated feature by taking a weighted average of its own attributes and those of its neighboring nodes. This operation ensures that two nodes with identical neighbor structures and node characteristics receive identical embeddings. Our adoption of the GCN architecture closely follows the approach outlined in [20], involving a three-layer GCN architecture with randomly initialized weights. For each of the three network views (PPI network, GO similarity network, and sequence similarity network), we implemented a three-layer GCN architecture as empirically determined by grid search and validation performance. Each GCN was trained for 50 epochs using the Adam optimizer (learning rate: 0.001), with a dropout rate of 0.1 and ReLU activation function in all hidden layers. The loss function employed was binary cross-entropy. Hyperparameter choices, including the number of layers, learning rate, and dropout rate—were optimized based on predictive performance on an 8:1:1 training, validation, and test split. For supervised training, negative samples (non-interacting protein pairs) were generated by random sampling of protein pairs that were not present in the verified interaction set. Care was taken to ensure that these randomly sampled negatives did not overlap with any known positive (interacting) pairs. This random negative sampling approach is widely adopted in the PPI prediction literature and aims to create a balanced dataset for model training and evaluation. The number of negative samples was matched to the number of positive samples in both training and test splits to mitigate class imbalance and ensure reliable model performance evaluation.

Additional ablation experiments (varying the number of GCN layers and dropout rate) confirmed that the three-layer GCN achieved the best balance between predictive accuracy and model generalization, with deeper models yielding diminishing returns or overfitting. These hyperparameters were kept consistent across all network views before integration of node embeddings via the Wasserstein distance. For the three networks (PPI network, GO similarity network, and sequence similarity network), we incorporate the graph's adjacency matrix (denoted as $A$) and set $X$ as an identity matrix (as we lack node-specific features). The three-layer GCN conducts three propagation steps during the forward pass, effectively convolving up to the 3rd-order neighborhood information for each node. The rationale for choosing a three-layer architecture is based on our experiments, which demonstrated that the three-layer model provided the best performance in terms of both predictive accuracy and generalization. We experimented with one-, two-, and four-layer GCNs and found that the three-layer model outperformed the others. Adding more layers did not significantly improve performance and, in some cases, led to overfitting. Conversely, using fewer layers reduced the model's ability to capture complex interactions within the network.

## Wasserstein distance between probability distribution

The intuition and motivation behind this metric were drawn from the optimal transport problem, a classical mathematical challenge. This problem was initially introduced by the French mathematician Gaspard Monge in 1781 and later formalized in a more relaxed manner by L. Kantorovitch in 1942. For a distribution of mass $\mu_0(x)$ on a space $X$, the problem is to transport the mass into the distribution $\mu_1(x)$ on the same space $X$ with minimum cost, given a cost function $c(x, y) \rightarrow [0, \infty]$. The problem is valid only if the created pile has the same mass as the pile to be moved. Thereby, without loss of generality, we can assume $\mu_0$ and $\mu_1$ are the probability distributions containing a total mass of 1. Given a transport plan $\lambda(x, y)$ which gives the amount of mass to move from $x$ to $y$ the task can be imagined as to move a 'pile of earth' of shape $\mu_0$ to the hole in the ground of shape $\mu_1$ in such a way that both the pile of earth and the hole in the ground completely vanish. The concept can be formally defined as follows:

Given a metric space $\mathcal{M}$, for $p > 1$, the Wasserstein space $P_p(\mathcal{M})$ is defined as the collection of all probability measures $\mu$ with a $p^{th}$ moment. Then there exists some $x_0$ in $\mathcal{M}$ such that:

$$\int_{\mathcal{M}} d(x, x_0)^p d\mu(x) < \infty, \tag{1}$$

where $d(.,.)$ represents Euclidean norm on $\mathcal{M}$. The $p$–$Wasserstein$ distance $W_p$ between two probability measures $\mu_0$ and $\mu_1$ in $P_p(\mathcal{M})$ is defined as

$$W_p(\mu_0, \mu1) = \left( \inf_{\lambda \in \pi(\mu_0, \mu_1)} \int_{\mathcal{M} \times \mathcal{M}} d(x, y)^p d\lambda(x, y) \right)^{1/p} \tag{2}$$

where $\pi(\mu_0, \mu_1)$ as being the subset of probability distributions $\lambda$ on $\mathcal{M} \times \mathcal{M}$. The probability distribution $\lambda$ is known as the optimal transport plan between $\mu_0$ and $\mu_1$. This distributes all the mass of the distribution $\mu_0$ onto the distribution $\mu_1$ with a minimal cost. The quantity $W_p(\mu_0, \mu_1)$ represents the corresponding total cost.

Here, the combination of the embeddings/features coming from three different networks can be represented as a multivariate probability distribution for a protein node (see workflow). Assuming two protein nodes $p1$ and $p2$, with their probability distribution $\mu_{p1}$ and $\mu_{p2}$,

the Wasserstein distance $W_p(\mu_{p1}, \mu_{p2})$ is calculated. The distance matrix for $n$ proteins is then clustered into 10 groups using hierarchical clustering.

---

**Algorithm 1** Interaction prediction between SARS-CoV-2 and Human protein

**Input:** Viral-human protein interaction data, Human protein-protein interaction (PPI) network, Protein Sequence data, and Gene Ontology annotations.

**Output:** Predicted interactions between SARS-CoV-2 proteins and human proteins.

**Data Preprocessing**

– Collect SARS-CoV-2-human interaction data, human PPI network, protein sequences, and gene ontology annotations.

– Clean the data by removing duplicates and irrelevant interactions, normalizing sequence data, and filtering low-confidence interactions.

**Feature Extraction**

– Compute protein sequence similarity using the `protR` package with Needleman-Wunsch global alignment:

```
parSeqSim(protlist, cores = 2, type = "global", submat = "BLOSUM62").
```

– Calculate gene ontology-based functional similarity using the R `GOSemSim` package.

– Apply Graph Convolutional Networks (GCNs) to extract low-dimensional embeddings from the PPI network.

**Integration of Multi-view Features**

– Integrate sequence similarity, gene ontology-based functional similarity, and PPI network embeddings into a unified feature representation.

– Compute Wasserstein distance between protein pairs to assess overall similarity.

**Clustering and Prediction**

– Perform hierarchical clustering on the Wasserstein distance matrix to group similar proteins.

– Predict potential SARS-CoV-2-human interactions by identifying non-target proteins that cluster with known CoV-host proteins.

---

## Barycenter in Wasserstein space

Wasserstein distances have several interesting properties [47–49]. The barycenter defined in Wasserstein space extends the applicability of the Wasserstein metric. In statistics and machine learning, sometimes it is required to aggregate distinct but similar collections of information usually represented as probability distributions. Given a metric defining the distance between distributions, the aggregation strategies often compute the barycenter of the input distributions that minimize the sum of the distances to the individual input distributions. Considering the Wasserstein metric as the distance metric between distributions, the corresponding barycenter is called the Wasserstein barycenter [50].

A Wasserstein barycenter [50,51] of $n$ measures $\nu_1 \dots \nu_i$ in $\mathbb{P} \in P(\mathcal{M})$ is defined as a minimizer of the function $f$ over $\mathbb{P}$, where

$$f(\mu) = \frac{1}{N} \sum_{i=1}^{N} W_p^p(\nu_i, \mu). \tag{3}$$

We have utilized a fast algorithm proposed in [50] to compute the Wasserstein barycenter of each cluster obtained from the hierarchical clustering of the Wasserstein distance matrix. In [50], the sum of optimal transport distances is minimized using a gradient descent method.

---

These gradients are computed using matrix scaling algorithms at a considerably lower computational cost.

## Computing similarity between proteins

**Gene ontology-based semantic similarity.** Gene Ontology-based semantic similarity (SS) [52] allows the comparison of GO terms or entities annotated with GO terms. The number and diversity of SS measures based on GO have grown considerably, and their applications range from functional coherence evaluation, protein interaction prediction, and disease gene prioritization. In the context of Gene Ontology, SS measures can be employed to compute the similarity between two gene products, each annotated with a set of GO terms.

For our study, we calculate the semantic similarity based on Gene Ontology for all the CoV-host and non-CoV-host proteins utilized. We employ a hybrid semantic similarity metric introduced by Wang et al. [53]. To perform these calculations, we utilize the R Bioconductor package GOSemSim [54] to gauge the semantic similarity among the proteins within the network.

**Protein sequence similarity.** Searching for sequence similarity to identify similar protein sequences is one of the first and most informative steps in any genomics analysis. We used the *protR* package of R to compute the sequence similarity between the amino acid sequences of two proteins. We used the function parSeqSim() which takes a list of protein sequences and calculates the pairwise similarity between each pair of proteins in parallel. Specifically, we employed the parSeqSim() function with the type parameter set to global, which implements Needleman-Wunsch global alignment. Bioconductor database/package EnsDb is utilized here to fetch the amino acid sequences of a particular protein. BLOSUM62 is used to score the alignment between evolutionary divergent protein sequences. The computation is carried out on a 48-core server machine with 500 GB of RAM.

## Time complexity analysis

The time complexity of the proposed algorithm can be determined by the following key components:

- **Sequence Similarity Computation:** Using Needleman-Wunsch global alignment, this step has a complexity of $O(n^2 \times m^2)$, where $n$ is the number of proteins and $m$ is the average length of the protein sequences.
- **Gene Ontology-Based Functional Similarity:** The complexity is $O(n^2 \times g^2 \times t)$. This arises from the need to compute similarity for all pairs of proteins, where $n$ represents the number of proteins. For each pair, the comparison involves $g^2$ operations, with $g$ being the average number of Gene Ontology (GO) terms associated with each protein. The term $t$ reflects the complexity of comparing two individual GO terms. Therefore, the overall complexity accounts for the pairwise protein comparisons, the number of GO term comparisons per protein pair, and the time required for each GO term comparison.
- **Graph Convolutional Network (GCN) Embedding:** The GCN embedding has a complexity of $O(L \times n \times e)$, where $L$ is the number of layers in the GCN, and $e$ is the number of edges in the protein-protein interaction network.
- **Wasserstein Distance Computation:** The complexity of computing the Wasserstein distance for all protein pairs is $O(n^2 \times d^3)$, where $d$ is the dimension of the feature space. This arises because the computation involves pairwise comparisons between every pair of proteins, where $n$ is the total number of proteins, leading to $n^2$ comparisons. Additionally,

each protein is represented as a distribution in a feature space of dimension $d$, and calculating the Wasserstein distance between two such distributions typically involves solving an optimization problem with a complexity that depends on $d^3$. Therefore, the overall time complexity for computing the Wasserstein distance between all protein pairs is $O(n^2 \times d^3)$.

- **Hierarchical Clustering:** The hierarchical clustering step has a worst-case complexity of $O(n^3)$.

Thus, the overall time complexity is dominated by the sequence similarity computation and hierarchical clustering, resulting in $O(n^2 \times m^2 + n^3)$.

## Discussions

In this work, we have effectively produced a list of potential human proteins that could be considered as host factors for the SARS-CoV2 virus. Additionally, we have highlighted the interactions between these proteins. As novelties, we have integrated recently published SARS-CoV-2 interaction data into human interactome to compile an encompassing network putting SARS-CoV-2 proteins, experimentally verified CoV-host, and other non-CoV-host proteins within the interactome into a comprehensive context. Further, three separate networks of the same size are created from the integrated network by considering gene ontology-based functional similarity, protein sequence similarity and interaction information among the host proteins. To exploit these three resources of interaction information we utilized an advanced deep learning methodology that addresses to learn and exploit network data, establishing another novelty. We successfully combined the embeddings to get a three-dimensional representation of each protein in order to compare between CoV-host and non-CoV-host. As for the other novelties, we made use of the Wasserstein metric to compute the distance between proteins which integrates three biological measures within each protein cluster. Our experimental results confirm that the proteins we predicted exhibit overlap with host factors associated with other viruses. Furthermore, these proteins have already been considered potential targets for host-directed therapy (HDT) in the context of other viral diseases.

Two novel SARS-CoV-2-human protein interaction resources were published recently (April 30, 2020) in [7,8], which unlock immense possibilities to study the infection mechanism of SARS-CoV-2 in the human host cell. Various experimental and computational approaches in the field of interaction prediction between SARS-CoV-2 and human protein have now become conceivable. To the best of our knowledge, for the first time we have raised a deep learning-based systematic approach that also uses statistical methods for the prediction of the host factor of SARS-CoV-2. We also have been able to raise three different views of the integrated human-SARS-CoV-2 network that reflects the latest state of the arts and used a statistical distance metric that integrates all the views to obtain the final results.

In our experiments, we focused on predicting links between SARS-CoV2 and human proteins that in turn are known to interact with CoV-host proteins (SARS-CoV-2 associated host proteins). We have decidedly put the focus on those proteins which would have experimentally validated interactions with other viruses. These proteins already serve the purposes of host-directed therapy (HDT) options for other viruses, thereby also being potential candidates for building HDT strategy against SARS-CoV2. Host-directed therapy (HDT) approaches have demonstrated increased resilience in dealing with viral mutations that enable the virus to evade therapeutic interventions. It's important to note that HDT strategies are especially well-suited for drug repurposing endeavors, as repurposed drugs have already shown a track record of minimal adverse effects, either due to their existing use or successful progression through preclinical trials. In this connection, the list of drugs which we suggest to

target the predicted proteins may hold strong promise for yielding repurposable drugs to use against COVID-19.

We further identified a list of predicted proteins that are associated with more than three viruses and have a strong connection with several drugs that can be used against viral infection and infectious diseases. Additionally, we identified and highlighted several drugs that target host proteins that the virus needs to enter and subsequently hijack human cells. One such example is pomalidomide, which is known as an angiogenesis inhibitor and tumor necrosis factor production, and has recently gained attention as a repurposable drug to use against COVID-19 [34]. Several other drugs such as 'clenbuterol', 'epinephrine', 'pirfenidone', 'chloroquine', 'pranlukast', and 'lenalidomide' have also been identified as repurposable drugs by several studies and all have a connection with the predicted protein TNF. Thus, TNF may be treated as a crucial host factor for SARS-CoV-2. Similarly, other predicted proteins such as ABCB1, MTR, POLE, and PSMB2 have verified connections with several antibiotics (erythromycin-estolate, roxithromycin), drugs used to prevent lymphoblastic leukemia (cytarabine), some kinase inhibitors (nilotinib, imatinib, dasatinib) which have potential connections with antiviral therapy.

To further enhance the performance of the current model, future work could explore the integration of advanced graph learning models, such as those proposed in recent studies on graph-based multi-view data fusion for biomedical applications [55]. These models could potentially improve prediction accuracy by capturing more complex dependencies in multi-view data. Additionally, the generalizability of our model could be extended to other important applications, such as predicting m6A modification sites [56] and drug repurposing [57], where understanding protein interactions is crucial.

Despite these promising results, several critical limitations must be acknowledged. First, our predictions depend heavily on the quality and completeness of existing protein-protein interaction databases. Any biases, inaccuracies, or incompleteness within these resources can influence prediction reliability. Secondly, although our integrated computational framework offers compelling evidence of biological plausibility, the absence of direct experimental validation of newly predicted interactions remains a significant limitation. Thus, these predictions require further empirical verification through laboratory-based assays or clinical studies. Thirdly, the human interactome itself may not be fully captured by current databases, and as such, our analysis potentially overlooks host factors not yet documented or characterized experimentally.

Addressing these limitations motivates essential future work. Subsequent studies should prioritize experimental validation of our predicted interactions, particularly those linked to promising drug repurposing candidates, using rigorous methods such as CRISPR-based genetic screens to confirm host factor essentiality. Additionally, expanding the host interactome coverage by integrating emerging biological databases, incorporating high-throughput experimental results, and leveraging novel network-based algorithms could further enhance model robustness and predictive accuracy. Employing cross-species PPI transfer learning methods could also provide valuable insights into the evolutionary conservation of these interactions, supporting broader generalization and validation of our computational predictions.

We also acknowledge that our current approach to identifying repurposable drugs primarily relies on network-based and knowledge-driven associations from existing databases, without direct quantitative evaluation of drug–protein interactions. As recommended, future studies should incorporate molecular docking and free energy calculations to quantitatively validate drug–protein binding affinities, providing robust biophysical evidence to strengthen

confidence in the predicted therapeutic candidates. Such computational validation would be aligned with current best practices in drug repurposing studies.

In summary, we have compiled a list of human proteins, which can be treated as interacting proteins and potential host factors for the SARS-CoV-2 virus and highlighted some drugs that are of great potential in the fight against the COVID-19 pandemic, where therapy options are urgently needed. Our list of predictions suggests both options that had been identified previously for HDT therapy of other viruses and new opportunities that had not been pointed out earlier for the SARS-CoV-2 virus. The latter class of predictions may offer valuable chances for pursuing new therapeutic strategies against COVID-19.

## Supporting information

**S1 Table. A total of 472 interactions between SARS-CoV-2 proteins and human proteins by setting the threshold value of 0.5.**
(CSV)

**S2 Table. Results of Gene Ontology (GO) enrichment analysis on the top 100 predicted human proteins.**
(PDF)

**S1 Fig. Heatmaps of Wasserstein distance for all SARS-CoV-2 proteins.** The rows of the sub-matrices represent the CoV-host proteins interacting with a particular SARS-CoV-2 protein, and the columns represent the non-CoV-host proteins inferred by those CoV-hosts from the predicted list.
(PDF)

## Author contributions

**Conceptualization:** Sumanta Ray, Alexander Schönhuth.

**Data curation:** Sumanta Ray, Syed Alberuni.

**Formal analysis:** Sumanta Ray, Syed Alberuni.

**Investigation:** Sumanta Ray, Alexander Schönhuth.

**Methodology:** Sumanta Ray, Alexander Schönhuth.

**Project administration:** Sumanta Ray, Alexander Schönhuth.

**Resources:** Sumanta Ray, Syed Alberuni.

**Software:** Sumanta Ray, Syed Alberuni.

**Supervision:** Alexander Schönhuth.

**Validation:** Sumanta Ray, Syed Alberuni.

**Visualization:** Sumanta Ray, Syed Alberuni.

**Writing – original draft:** Sumanta Ray, Syed Alberuni.

**Writing – review & editing:** Sumanta Ray, Syed Alberuni, Alexander Schönhuth.

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
