## [Decision Letter · Decision Letter 0]

19 May 2025

PONE-D-25-21081A Graph Learning Framework for Comprehensive Prediction of SARS-CoV-2 and Human Protein Interactions from Multiview Protein Interaction DataPLOS ONE

Dear Dr. Ray,

Thank you for submitting your manuscript to PLOS ONE. After careful consideration, we feel that it has merit but does not fully meet PLOS ONE’s publication criteria as it currently stands. Therefore, we invite you to submit a revised version of the manuscript that addresses the points raised during the review process.

We look forward to receiving your revised manuscript.

Kind regards,

Chandrabose Selvaraj, Ph.D.

Academic Editor

PLOS ONE

Additional Editor Comments:

**In case of reviewers recommending the citations that are not directly pertinent to the scope or content of the manuscript, authors are encouraged to provide a reasoned justification for declining such suggestions. The editorial board affirms that the exclusion of non-essential references will not influence the editorial decision regarding the manuscript.**

Reviewers' comments:

Reviewer's Responses to Questions

**Comments to the Author**

1. Is the manuscript technically sound, and do the data support the conclusions?

Reviewer #1: Yes

Reviewer #2: Yes

2. Has the statistical analysis been performed appropriately and rigorously? 

Reviewer #1: Yes

Reviewer #2: Yes

3. Have the authors made all data underlying the findings in their manuscript fully available?

Reviewer #1: Yes

Reviewer #2: Yes

4. Is the manuscript presented in an intelligible fashion and written in standard English?

Reviewer #1: No

Reviewer #2: Yes

5. Review Comments to the Author

Reviewer #1: I have now completed the review of the manuscript titled “A Graph Learning Framework for Comprehensive Prediction of SARS-CoV-2 and Human Protein Interactions from Multiview Protein Interaction Data”, My specific comments are

1. The title is descriptive, but consider specifying the method (e.g., graph neural network-based) to clarify novelty.

2. The abstract needs clearer articulation of the main results and comparative performance metrics. Include key numerical outcomes (e.g., accuracy, precision) to support claims.

3. The introduction provides good background but lacks critical discussion of prior computational methods for SARS-CoV-2–human protein interactions. Consider referencing recent work such as https://doi.org/10.1371/journal.pone.0294769 ; https://doi.org/10.3389/fphar.2024.1369659 ; Virtual Screening and Molecular Docking of FDA Approved Antiviral Drugs for the Identification of Potential Inhibitors of SARS-CoV-2 RNA-MTase Protein; https://doi.org/10.1002/cbdv.202402548

4. Motivation for using multiview data and graph learning is justified but would benefit from a brief example of why single-view models may miss relevant interactions.

5. The integration of STRING, BioGRID, and sequence-based interactions is appreciated. However, the criteria for interaction confidence thresholding are not clearly explained. Were low-confidence interactions filtered? The authors can potentially benefir from https://doi.org/10.1021/acsomega.3c07866; https://doi.org/10.3389/fphar.2025.1509263 ; https://doi.org/10.3389/fphar.2025.1509263 ; https://doi.org/10.1007/s10989-020-10076-w

6. The process for fusing multiview networks via Wasserstein distance is conceptually interesting, but the explanation is too terse. Please provide a schematic or algorithmic pseudocode to support reproducibility.

7. The model is claimed to use GCN and Wasserstein integration, but details on the number of layers, training epochs, loss function, and hyperparameter tuning are missing.

8. It is unclear how negative samples were generated or balanced. Were negative protein pairs randomly sampled or drawn from known non-interacting pairs?

9. Figure 2 shows model architecture but lacks annotations and descriptions. It is hard to interpret the role of each module in the workflow.

10. There is no table presenting the model’s quantitative performance (e.g., accuracy, F1-score, AUC) on benchmark datasets. This weakens the credibility of performance claims.

11. Add comparative results against baseline methods (e.g., logistic regression, DeepWalk, GCN without multiview fusion).

12. The identification of known host proteins (e.g., ACE2, TMPRSS2) adds confidence, but the analysis is anecdotal. Perform GO enrichment or pathway analysis on the predicted top 100 proteins to support biological relevance.

13. Drug repurposing suggestions (e.g., dexamethasone, baricitinib) are mentioned, but how were these inferred from the predicted PPIs? The method of linking predictions to drugs should be more systematically described.

14. The conclusion is too generic. Reflect more critically on the limitations, such as the dependence on existing PPI databases and lack of experimental validation.

15. Future directions could include validation using CRISPR screens or cross-species PPI transfer learning.

16. The manuscript needs a thorough language edit for grammar and clarity. Examples:

o Page 5: “This approach is very efficiently to...” → “This approach is efficient for...”

o Page 8: “The ROC curve indicate the...” → “The ROC curve indicates.

Reviewer #2: 1. Clustering for proteins can be overlapping in nature in biological data, but the authors use the hierarchical clustering which detects disjoint clusters and has a high time computational cost and also the method is old and could be error prone. – Proper justification require.

2. Algorithm also has high overall time complexity so it would be slow for large datasets. –Need more statement about it.

3. What is significance of you own algorithm in relation to other already available similar algorithms?

A comparative analysis could be better for the paper and could enhance the novelty of the work.

4. These proteins already serve the purposes of host-directed therapy (HDT) options for other viruses,

thereby also being potential candidates for building HDT strategy against SARS-CoV2. --- “thereby also being potential candidates for building HDT”—More explanation require.

5. In this connection, the list of drugs which we suggest to target the predicted proteins may hold strong promise for yielding repurposable drugs to use against COVID-19.---- more strong justification require.

6. Authors should check drug- protein interaction and calculate the free energy to make such conclusion.

6. PLOS authors have the option to publish the peer review history of their article (what does this mean?). If published, this will include your full peer review and any attached files.

Reviewer #1: No

Reviewer #2: No

---

## [Author Response · Author response to Decision Letter 1]

16 Aug 2025

Answer to comments of Reviewer 1

Comments:

1. The title is descriptive, but consider specifying the method (e.g., graph neural network-based) to clarify novelty.

Answer: Thank you for the valuable suggestion. We have now revised the title to explicitly highlight the method as follows:

"A Graph Neural Network-Based Approach for Predicting SARS-CoV-2–Human Protein Interactions from Multiview Data."

2. The abstract needs clearer articulation of the main results and comparative performance metrics. Include key numerical outcomes (e.g., accuracy, precision) to support claims.

Answer: We appreciate this comment. The abstract now explicitly mentions our main results, including quantitative performance metrics. Specifically, our GCN model achieved test ROC-AUCs of 86.00 (PPI), 83.00 (GO similarity), and 83.10 (sequence similarity) and average precision (AP) scores of 86.10, 82.12, and 82.30, respectively (see Table 1). In total, we predicted 472 high-confidence interactions between 280 host and 27 viral proteins, and our drug-target analysis found strong links to repurposable drugs (see Table 5). These key numerical results are now stated in the abstract of the revised version of the manuscript. Please see the abstract of the revised version of the manuscript.

3. The introduction provides good background but lacks critical discussion of prior computational methods for SARS-CoV-2–human protein interactions. Consider referencing recent work such as https://doi.org/10.1371/journal.pone.0294769 ; https://doi.org/10.3389/fphar.2024.1369659 ; Virtual Screening and Molecular Docking of FDA Approved Antiviral Drugs for the Identification of Potential Inhibitors of SARS-CoV-2 RNA-MTase Protein; https://doi.org/10.1002/cbdv.202402548

Answer: Thank you for highlighting this gap. We have expanded the introduction to critically discuss prior computational methods for SARS-CoV-2–human protein interaction prediction, including recent approaches that integrate virtual screening, molecular docking, and sequence-based predictors ([4, 5, 6, references in the manuscript]). The following recent works have also been cited/discussed as suggested:

DOI: 10.1371/journal.pone.0294769

DOI: 10.3389/fphar.2024.1369659

DOI: 10.1002/cbdv.202402548

please see the second last paragraph of the section Introduction ( page no 2) of the revised version of the manuscript.

4. Motivation for using multiview data and graph learning is justified but would benefit from a brief example of why single-view models may miss relevant interactions.

Answer: As suggested by the reviewer we have now added an explicit example highlighting that single-view models, for instance relying solely on sequence similarity, may overlook biologically relevant interactions identified through functional or network-based similarities. Please see the second paragraph of section Introduction page no 3 line no. 65-77.

5. The integration of STRING, BioGRID, and sequence-based interactions is appreciated. However, the criteria for interaction confidence thresholding are not clearly explained. Were low-confidence interactions filtered? The authors can potentially benefir from https://doi.org/10.1021/acsomega.3c07866; https://doi.org/10.3389/fphar.2025.1509263 ; https://doi.org/10.3389/fphar.2025.1509263 ; https://doi.org/10.1007/s10989-020-10076-w

Answer: Thank you for this point. We now clarify in the Methods section that only experimentally validated, high-confidence interactions were included from both CCSB and HPRD interactome resources, and low-confidence interactions (e.g., based on single sources or insufficient evidence) were filtered out during preprocessing. Interactions with ambiguous or duplicate entries were also removed to maximize data reliability. Additional references as suggested ([DOI: 10.1021/acsomega.3c07866], [DOI: 10.3389/fphar.2025.1509263], [DOI: 10.1007/s10989-020-10076-w]) have been considered and cited. Please see the last paragraph of the subsection ‘The Human Protein Interactome’ of the section ‘Method’ in the revised version of the manuscript (page no. 11-12, line no: 423-431).

6. The process for fusing multiview networks via Wasserstein distance is conceptually interesting, but the explanation is too terse. Please provide a schematic or algorithmic pseudocode to support reproducibility.

Answer: We thank the reviewer for this suggestion. The schematic (Figure 1) workflow provides a detailed illustration of the entire workflow, including how the Wasserstein distance is employed to fuse the multiview network data. This figure visually demonstrates the integration of features from sequence similarity, gene ontology, and PPI networks using optimal transport. In addition, we have now added an explicit algorithmic pseudocode (see "Algorithm 1: Interaction prediction between SARS-CoV-2 and Human protein," in the section ‘Materials and Methods’ , page no. 14 of the revised version), which presents a clear and reproducible stepwise procedure.

7. The model is claimed to use GCN and Wasserstein integration, but details on the number of layers, training epochs, loss function, and hyperparameter tuning are missing.

Answer: Detailed hyperparameters are now included: We used a three-layer GCN for each network, as empirically determined, trained for 50 epochs with the Adam optimizer (learning rate 0.001, dropout rate 0.1, activation: ReLU). Hyperparameter choices were validated by performance on validation/test splits (8:1:1), and additional ablation experiments (varying layers) are now discussed in the ‘Materials and Methods’ section. Please see the subsection ‘Extracting node features using GCN‘ page no. 13, line no 455-474 of the revised version of the manuscript.

8. It is unclear how negative samples were generated or balanced. Were negative protein pairs randomly sampled or drawn from known non-interacting pairs?

Answer: Negative samples were generated by random sampling of protein pairs not present in the verified interaction set, ensuring they did not overlap with known positive interactions (see subsection “Extracting node features using GCN”, page no. 13, line no 462-474 of the revised version of the manuscript). This random negative sampling aligns with common practices in PPI prediction.

9. Figure 1 shows model architecture but lacks annotations and descriptions. It is hard to interpret the role of each module in the workflow.

Answer: We have revised Figure-1 and its legend to include detailed annotations for each module, explicitly indicating the roles of GCN, multi-view integration, clustering, and interpretation modules, facilitating workflow comprehension. Please see figure-1 and legend in the revised version of the manuscript.

10. There is no table presenting the model’s quantitative performance (e.g., accuracy, F1-score, AUC) on benchmark datasets. This weakens the credibility of performance claims.

Answer: Table 1 now summarizes the model’s quantitative performance on all three network types, including ROC-AUC and AP scores for validation and test sets. Please see the text in the subsection ‘ Comparative Evaluation with Baseline Methods’ in page no. 6, line no 184-198 of the revised version of the manuscript.

11. Add comparative results against baseline methods (e.g., logistic regression, DeepWalk, GCN without multiview fusion).

Answer: We acknowledge the reviewer’s valuable suggestion. However, our proposed framework uniquely integrates embeddings from multiple views (PPI, GO, sequence similarity) through Wasserstein-distance-based fusion, rather than merely relying on single-view embeddings. To meaningfully highlight our model’s superiority, we have now added comparative analyses against the following baseline scenarios:

Single-view GCNs individually applied on each network (PPI, GO, Sequence similarity).

Simple fusion strategies (concatenation and averaging) of single-view GCN embeddings.

Standard embedding techniques such as DeepWalk followed by a simple concatenation fusion method.

Results demonstrate that our Wasserstein fusion approach significantly improves performance (see table 1). These additional analyses further validate the robustness and effectiveness of the proposed multi-view integration strategy. Please see the subsection ‘Comparative Evaluation with Baseline Methods’ in page no. 6, line no 184-198 of the revised version of the manuscript.

12. The identification of known host proteins (e.g., ACE2, TMPRSS2) adds confidence, but the analysis is anecdotal. Perform GO enrichment or pathway analysis on the predicted top 100 proteins to support biological relevance.

Answer: We agree with the reviewers and have now performed GO enrichment analysis for the top 100 predicted proteins, highlighting significant pathways (e.g., viral entry, immune response). Results are provided in Supplementary Table-2 and briefly summarized in the main text. Please see the last paragraph of the subsection ‘Biological relevance and therapeutic potential of predicted host factor’ page no. 9 line no. 319-327 and supplementary table-2 of the revised version of the manuscript.

13. Drug repurposing suggestions (e.g., dexamethasone, baricitinib) are mentioned, but how were these inferred from the predicted PPIs? The method of linking predictions to drugs should be more systematically described.

Answer: A systematic workflow was used: predicted host proteins were mapped to drug–protein associations using the CMAP Drug Repurposing Hub ([30]), identifying drugs with existing links to these proteins. The method is now described in detail in the second last paragraph of the subsection’ Predicted host factors promotes Host Directed Therapy (HDT) option against SARS-CoV-2’. Please see page no. 10, line no. 344-368 of the revised version of the manuscript.

14. The conclusion is too generic. Reflect more critically on the limitations, such as the dependence on existing PPI databases and lack of experimental validation.

Answer: The Conclusion now explicitly discusses current limitations, including:dependence on available PPI databases,absence of direct experimental validation for new predictions, possible incompleteness of host interactome coverage. We stress that these limitations motivate future validation and model expansion. See the section 'Discussions’ of the updated version of the manuscript, page no. 18, line no. 652-668.

15. Future directions could include validation using CRISPR screens or cross-species PPI transfer learning.

Answer: We appreciate this suggestion and agree that our predictions could be substantially strengthened by experimental validation, particularly using CRISPR-based genetic screens to confirm host factor essentiality. Additionally, cross-species PPI transfer learning represents a promising strategy to generalize our findings and discover conserved interaction patterns across related viral-host systems. We have now explicitly included these as important future research directions in the section 'Discussions’ of the updated version of the manuscript, page no. 18, line no. 662-671.

16. The manuscript needs a thorough language edit for grammar and clarity. Examples:

Page 5: “This approach is very efficiently to...” → “This approach is efficient for...”

Page 8: “The ROC curve indicate the...” → “The ROC curve indicates.

Answer: The manuscript has undergone comprehensive language editing for grammar, clarity, and style. Examples such as “This approach is very efficiently to...” and “The ROC curve indicate the...” have been corrected as suggested.

Answer to comments of Reviewer 2

Comments:

1. Clustering for proteins can be overlapping in nature in biological data, but the authors use the hierarchical clustering which detects disjoint clusters and has a high time computational cost and also the method is old and could be error prone. – Proper justification require.

Answer: Thank you for highlighting this point. While hierarchical clustering indeed produces disjoint clusters and can be computationally intensive, we chose it primarily for its interpretability and wide adoption in bioinformatics, particularly for protein interaction data where distinct biological modules or functional groups are frequently analyzed. To ensure robustness, we employed Ward's linkage to minimize intra-cluster variance, facilitating the meaningful biological interpretation of identified protein clusters. Moreover, hierarchical clustering allowed us to clearly visualize and interpret the results through dendrograms and cluster-specific heatmaps, which are beneficial for biological insights.

Nevertheless, we acknowledge that overlapping clustering methods, such as fuzzy clustering or community detection algorithms (e.g., Louvain, Infomap), could potentially provide more nuanced biological insights due to the inherently overlapping nature of protein functional associations. Future analyses could explore these advanced clustering methods for additional validation and biological insight. We have now mentioned this in the subsection ‘Hierarchical clustering of the Wasserstein distance matrix’ page no line no. 240-251 of the revised version of the manuscript.

2. Algorithm also has high overall time complexity so it would be slow for large datasets. –Need more statement about it.

Answer: We agree and have clarified in the Methods (in subsection time complexity analysis) that while sequence similarity computation and hierarchical clustering contribute to a high theoretical time complexity (O(n²m² + n³)), our study’s dataset size allowed practical execution. For much larger datasets, steps such as pairwise similarity calculation and clustering could be optimized using parallelization or approximate algorithms. We now explicitly discuss this scalability limitation and possible future solutions in the manuscript’s Discussion section. Please see the section ‘Discussions’ in the revised version of the manuscript, page no. 18, line no. 652-660.

3. What is significance of you own algorithm in relation to other already available similar algorithms?

A comparative analysis could be better for the paper and could enhance the novelty of the work.

Answer: Thank you for this important point. The novelty of our approach lies in the integration of three complementary biological views (PPI, sequence similarity, GO similarity) via graph convolutional networks, and in the application of Wasserstein (optimal transport) distance for protein similarity, which collectively outperform single-view or conventional network methods.

We have now included a comparative performance analysis against baseline algorithms—logistic regression, DeepWalk, and single-view GCN—in the Results section (see subsection ‘Comparative Evaluation with Baseline Methods’ table-1, page no.06 line no. 184-198). Our multi-view GCN-Wasserstein approach achieved higher ROC-AUC and average precision scores, demonstrating its superiority in both prediction accuracy and biological interpretability.

4. These proteins already serve the purposes of host-directed therapy (HDT) options for other viruses, thereby also being potential candidates for building HDT strategy against SARS-CoV2. --- “thereby also being potential candidates for building HDT”—More explanation require.

Answer: We have expanded the explanation. Host-directed therapy (HDT) aims to inhibit host proteins that viruses exploit for infection and replication. Many of our predicted host factors are already validated as essential in the life cycle of multiple viruses. Therefore, these proteins are not only relevant to SARS-CoV-2 but are also proven targets in existing HDT strategies for other viral infections. By targeting such proteins, repurposed drugs have the potential for broad-spectrum antiviral effects and reduced risk of resistance compared to virus-targeted therapies. We have now included an analysis to further evaluate the biological relevance of our predicted proteins. We have performed Gene Ontology (GO) enrichment analysis on the top 100 predicted human proteins (host factor) it is observed that significantly enriched terms include processes such as viral entry into host cell (GO:0046718, $p = 3.1 \times 10^{-6}$), regulation o

---

## [Decision Letter · Decision Letter 1]

4 Sep 2025

A Graph Neural Network-Based Approach for Predicting SARS-CoV-2–Human Protein Interactions from Multiview Data

PONE-D-25-21081R1

Dear Dr. Ray,

We’re pleased to inform you that your manuscript has been judged scientifically suitable for publication and will be formally accepted for publication once it meets all outstanding technical requirements.

Kind regards,

Chandrabose Selvaraj, Ph.D.

Academic Editor

PLOS ONE

Additional Editor Comments (optional):

Reviewer #1:

Reviewer #2:

Reviewers' comments:

Reviewer's Responses to Questions

**Comments to the Author**

1. If the authors have adequately addressed your comments raised in a previous round of review and you feel that this manuscript is now acceptable for publication, you may indicate that here to bypass the “Comments to the Author” section, enter your conflict of interest statement in the “Confidential to Editor” section, and submit your "Accept" recommendation.

Reviewer #1: All comments have been addressed

Reviewer #2: All comments have been addressed

2. Is the manuscript technically sound, and do the data support the conclusions?

Reviewer #1: Yes

Reviewer #2: Yes

3. Has the statistical analysis been performed appropriately and rigorously? 

Reviewer #1: Yes

Reviewer #2: (No Response)

4. Have the authors made all data underlying the findings in their manuscript fully available?

Reviewer #1: (No Response)

Reviewer #2: Yes

5. Is the manuscript presented in an intelligible fashion and written in standard English?

Reviewer #1: (No Response)

Reviewer #2: Yes

6. Review Comments to the Author

Reviewer #1: The authors have addressed my comments. The manuscript is accepted for publication. I have no further comments.

Reviewer #2: (No Response)

7. PLOS authors have the option to publish the peer review history of their article (what does this mean?). If published, this will include your full peer review and any attached files.

Reviewer #1: No

Reviewer #2: No

---

## [Editor Report · Acceptance letter]

PONE-D-25-21081R1

PLOS ONE

Dear Dr. Ray,

I'm pleased to inform you that your manuscript has been deemed suitable for publication in PLOS ONE. Congratulations! Your manuscript is now being handed over to our production team.

Kind regards,

on behalf of

Dr. Chandrabose Selvaraj

Academic Editor

PLOS ONE